# Medical Resource Use and Medical Costs for Radiotherapy-Related Adverse Effects: A Systematic Review

**DOI:** 10.3390/cancers14102444

**Published:** 2022-05-16

**Authors:** Yi Hsuan Chen, Dominique Molenaar, Carin A. Uyl-de Groot, Marco van Vulpen, Hedwig M. Blommestein

**Affiliations:** 1Erasmus School of Health Policy and Management, Erasmus University Rotterdam, 3062 PA Rotterdam, The Netherlands; uyl@eshpm.eur.nl (C.A.U.-d.G.); blommestein@eshpm.eur.nl (H.M.B.); 2Department of Otolaryngology-Head and Neck Surgery, Amsterdam University Medical Center (UMC), Vrije Universiteit Amsterdam, 1081 HV Amsterdam, The Netherlands; d.molenaar@amsterdamumc.nl; 3Holland Proton Therapy Center, 2629 JH Delft, The Netherlands; m.van.vulpen@hollandptc.nl

**Keywords:** radiotherapy, adverse effect, health resource, health care cost

## Abstract

**Simple Summary:**

Cancer patients who receive radiotherapy often suffer from adverse effects that require healthcare resources to manage. This study summarized evidence of healthcare resource use and costs related to radiotherapy-induced adverse effects and provided recommendations for including this evidence in economic evaluations. Our findings revealed unignorable differences for the same adverse effects, which implied that the potential for the economic burden of adverse effects was overestimated or underestimated.

**Abstract:**

Background: Despite the need for a proper economic evaluation of new radiotherapies, the economic burden of radiotherapy-induced adverse effects remains unclear. A systematic review has been conducted to identify the existing evidence of healthcare resource use and costs related to radiotherapy-induced adverse effects and also to provide recommendations for including this evidence in economic evaluations. Methods: This systematic review of healthcare resource use and/or medical costs related to radiotherapy-induced adverse effects was performed up until 2020, focusing on patients with head and neck cancer, brain cancer, prostate cancer, eye cancer and breast cancer. Results: Resource use for treating the same adverse effects varied considerably across studies; for instance, the cost for mucositis ranged from USD 2949 to USD 17,244. This broad range could be related to differences in (1) severity of adverse effects in the study population, (2) study design, (3) cost estimation approach and (4) country and clinical practice. Conclusions: Our findings revealed unignorable differences for the same adverse effects, which implied that the potential for the economic burden of adverse effects was being overestimated or underestimated in economic evaluation for radiotherapy.

## 1. Introduction

Radiotherapy has been the pillar of cancer treatment for decades, providing cancer patients with a possible cure, extended survival and symptom relief [1]. Radiotherapy could also be added to other treatment options, such as chemotherapy and surgery, to remove and shrink the tumor and reduce the cancer recurrence rate [2,3].

Unfortunately, the efficient tumor elimination of radiotherapy is also accompanied by adverse effects. The normal tissue near the tumor site, which is also exposed to radiation during radiotherapy, can be damaged and can cause adverse effects [4,5,6]. While some adverse effects induced by radiotherapy are temporary, some of them could last for a long time or become permanent [7,8]. These severe, long-lasting adverse effects not only cause a considerable negative impact on patients’ life expectancy and quality of life, but also require time-consuming, resource-intensive and costly medical management [9,10,11].

To lower the incidence and severity of adverse effects, numerous new, high-price health technologies, such as proton therapy, are developed to replace conventional radiotherapy [12,13]. Several studies showed that patients receiving innovative radiotherapy treatments have a significantly lower risk of adverse effects compared to conventional treatments [14,15,16]. However, the cost of these new technologies could impose an economic burden on the healthcare system [17,18]. To preserve the sustainability of the healthcare system and optimize resource allocation, a comprehensive economic evaluation is essential [19,20]. 

An economic evaluation that aims to advise policymakers must be comprehensive and should include all costs. The costs should account for both expenditures and savings associated with the use of new technologies. For example, for new health technologies that aim to lower the incidence and severity of adverse effects, the cost savings for avoiding adverse effects ought to be taken into account. Several studies reported cost savings for avoiding RIAE, but a systematic review that summarizes the currently available evidence is currently lacking [21,22,23]. 

To the best of our knowledge, there is no up-to-date systematic review focusing on the medical costs related to all RIAE. Only one systematic review focusing on the economic burden of one type of adverse effect (i.e., xerostomia) was published in 2010 [24]. In that review, the authors concluded that there was no data on resource utilization related to xerostomia. Cost estimations could differ considerably as different definitions, scopes and methods impact the results [25,26]. Differences could result in a considerable effect on the conclusions of economic evaluations. Despite the importance of adverse effects in an economic evaluation, there is yet no consensus or standard guidance on how to incorporate treatment-related adverse effects into the economic evaluation of new technologies in radiotherapy [27]. Ignoring this information gap could risk embedding biases while incorporating adverse effects in an economic evaluation.

This review aims to identify and assess the currently available evidence on healthcare resource use and costs related to the adverse effects induced by radiotherapy. For this search, several divergent tumor indications were selected. Head and neck cancer is known for the frequent and severe side effects which occur early in time. Breast cancer is the most common cancer in women in which long-term cardiac and lung side effects are especially relevant. Prostate cancer is the most common cancer in men, for which the potential benefit of proton therapy is still under discussion. Brain cancer consists mainly of low-grade glioma in young patients with long-term side effects on, e.g., cognition, workability, etc. Eye cancer is a very rare cancer for which the reduction of side effects is clear, but the societal impact is unclear. The results of this review could help set health-technology research priorities for the future by providing insight into the economic burden. In addition, the synthesized evidence will be suitable as parameters for economic evaluations and may help ensure the unbiased evaluation of irradiation-related health technologies. Lastly, guidance on incorporating reported results in an economic evaluation is provided to reduce bias and improve consistency in future economic evaluations.

## 2. Materials and Methods

This systematic review followed the PRISMA guidelines to identify published studies that report healthcare resource use and healthcare cost related to RIAE in patients with HNC, breast cancer, brain cancer, eye cancer and prostate cancer. The complete study protocol is registered on PROSPERO (nr. CRD42020193256).

### 2.1. Database Search

Five electronic databases, MEDLINE, EMBASE, Web of Science, Cochrane and PsycINFO, were queried from their inception until March 2020, following the Preferred Reporting Items for Systematic Reviews and Meta-Analyses guidelines, adhering to the following key search elements: healthcare resource use (such as resource use, cost, hospitalization and specialist visit), radiotherapy (such as radiotherapy and irradiation), adverse effects (such as toxicity, adverse reaction), specific symptoms of radiotherapy-related adverse effects (such as xerostomia, dysgeusia and oral mucositis) and the indications for radiotherapy (head and neck cancer, breast cancer, brain cancer, eye cancer and prostate cancer).

The search was restricted to English language publications. No restriction was applied to published year. The search strategy used is available as a Appendix A. Duplicate articles were removed by the Endnote function.

### 2.2. Study Selection

The identified titles and abstracts were screened for eligibility by two independent reviewers, and full texts of relevant citations were obtained. To avoid missing any relevant literature, full texts were also obtained and screened for articles that did not provide sufficient information in their abstract. References of previous (systematic) reviews and primary literature were screened as well. Two reviewers screened and reviewed the full text independently, with disagreements resolved by discussion.

The inclusion criteria were: (1) studies that report healthcare resource use or medical cost for RIAE management; (2) studies that defined their study population as patients with head and neck cancer, breast cancer, brain cancer, eye cancer or prostate cancer. As for study design, cross-sectional, case-control or cohort studies that report healthcare resource use or healthcare cost for radiotherapy-related adverse effects were included. Randomized clinical trials (RCT) reporting on interventions that aim to manage the severity of RIAE were also included. In these RCTs, the outcome could consist of healthcare resource use or healthcare costs related to RIAE. 

The exclusion criteria were: (1) studies that did not specify resource use or costs related to RIAE; (2) studies that reported resource use or costs at an aggregated level, such as reporting only the total cost of the entire treatment course without detailed information on RIAE; (3) modeling-based studies that employed input parameters from other sources, if these sources were already included; (4) RCTs that focus on interventions for RIAE diagnosis. Furthermore, review articles, editorials, letters and abstracts for conferences were omitted as well.

### 2.3. Data Extraction

From the eligible studies, the following data were extracted; the general study characteristics (e.g., first author, year of publication, country of study), details of the study design (e.g., study type, cancer type, adverse effect type, the severity of the adverse effect and follow-up time), information on healthcare resource use (e.g., healthcare resource type, the amount of resource use), and healthcare costs. Two reviewers extracted data independently, with doubts resolved by discussion.

### 2.4. Reporting Outcome and Statistical Analysis

Firstly, the general study characteristics (e.g., year of publication, country, cancer site, adverse effect type, study design) of included literature were summarized. 

Secondly, the reported healthcare resource use related to RIAE was obtained and converted to monetary terms. For instance, RIAE may lead to gastrointestinal tract hemorrhage, and its treatment is associated with additional admission days. These additional admission days were then converted into monetary terms using the National Health Service (NHS) price set (2016–2017). The NHS price set is a list of national tariffs (average costs) used by healthcare providers in the United Kingdom to provide care efficiently. The healthcare costs were calculated by multiplying the unit price of healthcare services with the resource use of health care. Using one public price set to calculate the costs could increase the comparability between studies by eliminating the variation from healthcare price differences among different countries. 

Thirdly, for studies with costs as an outcome, the cost results were extracted. All costs, including the reported healthcare cost and the cost calculated from reported resource use, were translated to U.S. dollars with the annual average ratio of the published year of the study [28]. The costs were adjusted for inflation to 2020 with the CPI rate of the country where the study took place. The reported or calculated costs, along with the healthcare resource type, were then grouped according to the adverse effect type and study type. 

Lastly, the costs, reported or calculated, related to the same adverse effect were compared. The methods used to categorize the severity of adverse effects and the characteristic of the study population were studied as well. 

The average economic impact of each adverse effect estimated from modeling studies or non-modeling studies were listed parallelly.

## 3. Results

### 3.1. Database Search

The flow of studies identified, screened, excluded and included is shown in the form of a PRISMA chart (Figure 1). Initially, 2939 studies were identified after de-duplication; 2733 studies were excluded based on title and abstract screening; 206 full-text studies were reviewed; and 31 studies fulfilled the inclusion criteria and were included for data extraction. 

### 3.2. Study Characteristics

Table 1 presents the study characteristics of the 31 included studies. Most studies were published in the U.S. (58.1%), with twelve non-modeling-based studies, e.g., clinical trials and observational studies, and six modeling studies, e.g., cost-effectiveness analysis. Japanese and Dutch studies each encompassed 9.4% of the total included studies. Head and neck cancer (HNC) (72.2%) was the most common cancer type studied within the non-modeling-based studies, while prostate cancer (54%) was the most frequently studied cancer type within modeling studies. 

The number of studies focusing on the other three cancer types was relatively small (N = 5, 16%) (Table 2). In non-modeling-based studies for HNC, mucositis was the most frequently studied RIAEs (8/14). In prostate cancer studies, GI toxicity was included in all studies, while GU toxicity was included in 42% (4/11) of the studies. Non-modeling-based studies for brain cancer, breast cancer or eye cancer were scarce or unavailable. 

Of the included studies, 94% (29/31) used a payer perspective, which included costs covered by a healthcare insurance company or national healthcare plan. Out-of-pocket expenses and productivity costs were not included. One study, Schnur et al. [29], took the health care sector perspective, which included patients’ out-of-pocket expenses. Only one study, Lundkvist et al. [30], took a societal perspective and included patients’ productivity loss due to toxicities.

Of the studies with cost as their outcome, six estimated the institutional cost, while three studies reported costs based on reimbursement tariffs. Schnur et al. [29] examined out-of-pocket expenses estimated by a questionnaire-based survey. 

### 3.3. Healthcare Resource Use

Five studies addressed resource use related to oral mucositis, and the most frequently included healthcare resources were hospitalization days and nutrition supports (e.g., dietician consultation, feeding tube installation and nutrition supplements). However, the duration of hospitalization varied from 0.5 to 47 days. Elting et al. [31] and Murphy et al. [32] both stated similar duration of hospitalization (range from 0.5 to 2 days), while Kubota et al. mentioned 47 days.

The costs calculated from the reported resource use are listed in Table 3 (in USD). The cost calculations of similar adverse effects (e.g., mucositis) included different types of resource use and varied from USD 744 to 15,986. In Peterman et al. [33], the duration of hospitalizations and medication use were not indicated, while Elting et al. [31] reported the duration of hospitalizations, gastrostomy tube use and medication use for pain control. Less variation was observed for studies including the same resource use categories (e.g., Altman et al. [34] and Bennett et al. [35]). The adverse effect with the highest resource use was mucositis, for which the longest hospitalized duration was reported.

### 3.4. Healthcare Cost

Table 4 lists the results of the included studies that reported RIAE-related healthcare costs. The cost for the same adverse effect could differ up to five times. For example, the cost related to mucositis ranged from USD 4312 to 20,728. Variations could be related to differences in the inclusion of cost categories and the databases. Due to the different resource use categories included in each study, the healthcare costs (calculated from resource use) varied across studies. In Redmond et al. [43], aggregated cost data of several healthcare services (e.g., inpatient care, procedures, blood transfusion, etc.) were obtained and provided a more comprehensive result. Meanwhile, Voong et al. [44] estimated this cost based on medicine and procedure expense data.

For most RIAE-related costs, the main cost drivers were the surgeries or invasive procedures, such as sigmoidoscopy for GI adverse effects and percutaneous nephrostomy tube placement for radiation cystitis. The average duration of hospitalization for radiation cystitis was four days, yet plenty of invasive procedures were required. As a result, the cost related to radiation cystitis was higher than other RIAEs.

A comparison of the calculated costs from resource use and reported costs by the study showed that the latter was higher. The differences could range from 1.4 times (Peterman et al. [33]) to 1.7 times (Elting et al. [31]). The category differences in healthcare resource types were observed. In studies that reported resource use, the included resource types were more limited than the studies that reported costs. Even in studies dealing with both resource use and costs, the reported costs were calculated according to a more comprehensive resource type list than the reported resource use. For instance, Peterman et al. [33] indicated only the used numbers of outpatient care and nutrition supports, while its reported costs consist of the expenses from hospitalization, outpatient care, nutrition supports and medication costs. 

### 3.5. Modeling Studies

Table 5 lists the results extracted from the modeling studies. The costs related to GI toxicity raised as its severity increased. The reported GI toxicity-related costs varied from USD 47 to 1054 for grade 2 and from USD 2108 to 5678 for grade 3. 

For studies that focused on medulloblastoma, the experts’ ideas on the costs related to hearing loss could range from USD 2360 to 5829. The cost related to hearing loss was USD 2360 according to Hirano et al. [55], which included the cost of hearing aids, annual hearing tests for the following two years and hearing aid fitting test; whereas, Lundkvist et al. [30] reported the costs at USD 5829 per year. On the other hand, the costs for treating growth hormone deficiency (GHD) were similar between studies. 

Most of the modeling-based studies have gained insight into the healthcare resource use related to RIAE from healthcare experts’ inputs and empirical data. These inputs could depend on their perceived clinical practice, experience and preferences. The nature of this approach could partly explain why studies conducted within the same country shared a more similar idea of the costs (Vanneste et al. [51] and Van Wijk et al. [52]). 

From modeling studies that reported GI and GU toxicity, different methods were used to introduce adverse effect-related costs into their models. Some studies, such as Van Wijk et al. [52], Vanneste et al. [51] and Yong et al. [22], distinguished different levels of severity for adverse effects and assigned associated costs to these levels in their model, while others applied a pooled value.

## 4. Discussion

This study systemically reviewed and assessed existing evidence of RIAE-related healthcare resource use and medical costs for head and neck cancer, breast cancer, brain cancer, eye cancer and prostate cancer patients. It is, to the best of our knowledge, the first study to summarize and assess the currently available evidence that investigated differences in cost estimations of similar RIAEs and that examined the coherence of results estimated by different study designs.

The first finding of this study was the unevenly distributed research attention. The number of non-modeling-based studies reveals that the research attention was highly concentrated on HNC, especially the oral complications (including mucositis). This phenomenon was most likely related to the incidence and severity of oral complications. The incidence of radiotherapy-induced oral complications was relatively high; more than 80% of head and neck cancer patients who received radiotherapy suffered from it [59]. The symptoms of oral complications encompass consistent pain, a higher risk of infection and difficulty swallowing, potentially jeopardizing patient recovery.

In addition, it takes a long time for oral complications caused by radiotherapy to diminish, and no medical treatment can effectively manage them [59]. These characteristics of oral complications considerably impact patients’ quality of life. Consequently, this oral complication-related disutility drew the attention of clinical researchers. The established causal correlation between radiotherapy and oral complications could also be another reason behind this concentrated tendency. Unlike diarrhea, abdominal pain and nausea—which could result from a list of causes (e.g., infection, food poisoning, emotion)—oral complications are more specific and less susceptible to type one error. Nevertheless, there are RIAEs of which the incidence and severity could be lower by new technologies, but their costs are currently unknown. This lack of information could limit or restrict the economic evaluations of those new technologies.

Our results also highlighted the lack of information on other RIAEs and indications (i.e., brain cancer and breast cancer). This might mean that, so far, economic evaluations did not incorporate proper cost estimates for such RIAEs. Consequently, this introduces uncertainties into the economic evaluations and could increase the risk of suboptimal policy decisions. 

### 4.1. Health Care Resource Use

Eleven studies documented the healthcare resource use among HNC patients who underwent RIAE. Within those studies that reported resource use related to oral mucositis, the most frequently declared healthcare services were hospital days and nutrition supports. However, a considerable variation was observed among these studies in the duration of hospitalization. Some studies reported a similar duration of hospitalization (ranging from 0.5 day (Elting et al. [31]) to 2 days (Murphy et al. [32])), while Kubota et al. [36] reported 47 days. This difference could result from the clinical practice difference between countries or the hospitals’ policies. The Elting et al. [31] and Murphy et al. [32] studies both took place in the U.S., and the treatment routine for mucositis mainly encompassed the outpatient visit and dietician consultation. Especially Elting et al. [31], who declared that the study took place at the moment the hospital was running at its maximum capacity and that there was a disincentive for inpatient care. On the contrary, Kubota et al. [36] conducted the trial in Japan with a clear goal to discharge the patient from the hospital only after mucositis was resolved and the patient showed capability in oral ingestion. Consequently, due to the significant difference in hospital days, the healthcare cost calculated by Kubota et al. [36] was five times higher than Elting et al. [31]. In addition to the variation in clinical practice, the study design could impact its outcome. The research questions set by Elting et al. [31], Peterman et al. and Murphy et al. [32] were the resource utilization and cost related to mucositis. In contrast, Kubota et al. [36] and Tsujimoto et al. [37] were designed as clinical trials to estimate the efficacy of new treatment options for mucositis. As clinical trials do not reflect real-world situations, their results should be interpreted with caution since both overestimations (more intense monitoring) and underestimations (only surrogate outcomes were recorded) were possible [60]. To gain a reliable understanding of resource utilization, a non-interventional study in the same country is essential.

The healthcare resource use remained relatively consistent over time. With nine years difference, Altman et al. [34] and Bennett et al. [35] reported similar hospital days related to dysphagia; moreover, Peterman et al. [33] and Elting et al. [31] also showed similar outcomes despite a six-year gap. This consistency could imply that the treatment for either dysphagia or mucositis had not changed between 2001 and 2010. However, in 2007, the treatment guideline for mucositis had a major update by adding new treatment options (e.g., Palifermin), which could impact the healthcare utilities and costs that were not captured in the abovementioned studies. 

### 4.2. Health Care Costs

Nine studies declared healthcare costs related to RIAE. Among these studies, three of them also presented healthcare resource use. The calculated costs of resource use were mostly lower compared to those reported (from the same study). The differences could range from 1.4 times (Peterman et al. [33]) to 1.7 times (Elting et al. [31]), which could result from the different unit prices used in the cost calculation. In this study, the NHS pricing was used to calculate costs from resource use, and these prices might be below the U.S. prices. The other reason behind the higher reported costs was that not every healthcare resource component included in the cost calculation was described in the study. In Lang et al. [38], resource components, such as inpatient care, outpatient care, physician time and other health care, were included in the calculation. However, resource use was only reported for the main cost driver being the duration of inpatient care. Inpatient care accounted for 63% of the cost, and therefore, calculated costs based on resource use underestimate the total costs for this study.

Among the studies that focus on healthcare costs, one study, Nonzee et al. [45], showed much higher costs. Nonzee et al. [45] enrolled patients from medical centers and identified patients that were diagnosed with mucositis. Even though information on the severity of mucositis was not available, throughout the article, the author referred to them as “patients with severe mucositis”, which could imply that the patient population in this study was under more severe conditions than the study population of others. For instance, Peterman et al. [33] included patients who completed radiotherapy in the institution and were followed up for at least 12 months. With this approach, patients with mild conditions could be enrolled as well, and as a consequence, the average cost per patient might be lower. Our study emphasized the need for proper documentation of patient characteristics in costing studies. 

Comparing the cost parameters in modeling-based studies with the cost estimated in non-modeling-based studies, the estimated costs were much higher than the cost parameters. For instance, the estimated cost related to dysphagia was USD 2719 per year (Altman et al. [34]), whereas the cost parameter introduced by Ramaekers et al. [48] was USD 760. The number of studies already revealed the information gap, but the lack of information seems even more problematic. For brain cancer, medulloblastoma and breast cancer, no evidence of RIAE was available from non-model-based studies. Within those cancer types that had more evidence available, the information gap remained unignorable. Due to the low external validity, the costs reported from other countries were not necessarily suitable for the domestic situation [61]. The healthcare resource use and its cost rise as the severity of RIAE increases, which increases the need for using a consistent definition for the RIAE severity. 

There were some limitations to our study. One limitation was the selection of tumor sites instead of describing the entire range of proton therapy indications. The current use of proton therapy globally is very broad and shows a large heterogeneity [62]. Furthermore, literature on outcome, especially with regard to RIAE, is very scarce. We, therefore, chose not to investigate a heterogeneous mixture but focus on different tumor types, which differ in the amount of toxicity, acute or late toxicity, or are very frequent in rare tumors.

The other limitation lies in the cost calculation. The healthcare resource use was converted into monetary value by the NHS traffic price. This price set has an inherent difference from the actual cost perceived by the healthcare providers. This difference could not be adjusted in this review due to data limitations; the information on the actual costs per hospital was not available. This limitation indicated that the evidence from a domestic study would be preferable. The reason for this preference is that the price difference could be constrained when regulated by the same healthcare system. 

### 4.3. Guidance for Including Resource Uses and Costs into Economic Evaluation

Based on the findings of this study, guidance for integrating RIAE-related costs into the economic evaluation was developed. 

**Recommendation** **1.***Observational studies are the preferred research design for obtaining resource use to represent clinical practice*. 

The study design could impact the outcomes, and while RCTs ensure unbiased estimates for efficacy, their controlled circumstances cause (cost) findings not to be generalizable for clinical practice. The similarities between the patient characteristics and the context in trials and clinical practice determine how accurate actual costs are reflected in a study. The potential difference in clinical practice compared to trials make observational studies more preferable for estimating the costs of RIAE than clinical trials. 

**Recommendation** **2.***Accounting for the costs of different levels of severity of RIAE is necessary to appropriately include the economic burden of RIAE in an economic evaluation*. 

The characteristics of the study population have a considerable effect on the costs. The patients’ age and gender were often similar due to the prevalence of cancer, but the RIAE severity of the study population could vary across different studies. Compared to patients with severe RIAE, patients with mild (grade 1) RIAE were less likely to seek medical help or be diagnosed. This fundamental difference stressed the importance of elucidating the costs of different levels of severity of RIAE. This process is crucial for an accurate-estimated economic burden of RIAE. 

**Recommendation** **3.***To facilitate comparability, grading the severity with Radiation Therapy Oncology Group (RTOG) or the common terminology criteria for adverse events (CTCAE) is recommended*.

In order to facilitate the comparability between studies and applicability for future studies, grading the severity with commonly used criteria, such as Radiation Therapy Oncology Group (RTOG), and the common terminology criteria for adverse events (CTCAE) is highly recommended. 

**Recommendation** **4.***To ensure the economic burden of RIAE is accurately estimated, parameters from a domestic study are preferred. If not feasible, parameters adjusted to meet the domestic clinical practice could be an alternative*.

Nevertheless, the domestic country’s health service price, clinical practice and environmental factors should be considered. The saturation of hospital capacity and the medical dispute could influence healthcare providers’ preference between inpatient and outpatient settings [63]. Environmental factors such as health insurance policy and the national economic situation might suppress patients’ healthcare-seeking behavior. When results from studies conducted in other countries are used or significant time differences exist, the abovementioned factors should be considered. When conducting an economic evaluation, parameters from a study performed in the same country are highly preferable. If the domestic study was not available, adjusting the published evidence of resource use and costs to fit the domestic practice could also be an alternative. To make an appropriate adjustment, the researchers could make use of treatment guidelines, clinical experts’ opinions and local healthcare service prices.

**Recommendation** **5.***A bottom-up approach is preferred as it could increase transparency and use for future research*.

Lastly, the method for cost estimation, top-down or bottom-up approach, could directly influence to what extent healthcare resources were included. When conducting a study to estimate the economic impact of RIAE, a bottom-up approach could provide higher accuracy and more detail on the specific resource used. This detailed information on healthcare resource use was particularly valuable for international studies and studies conducted in other countries. 

Nevertheless, the accuracy of this approach highly relies on whether all RIAE-related healthcare services were included. If the researcher failed to include all, or sufficient, services, the results were underestimated. To avoid omitting those, using both treatment guidelines and interviewing healthcare providers or experts are recommended.

The guidance proposed in this study provides recommendations for conducting economic evaluations and for research focusing on the resource use or costs related to RIAE. An empirical study that reports outcomes (resource use or costs) for each severity group diagnosed with CTCAE or RTOG criteria of RIAE could facilitate the use of their results. Reporting healthcare resource use, instead of gross healthcare costs, could increase the transferability when using one’s results for economic evaluation in other geographic areas.

## 5. Conclusions

This review demonstrated the need for further research focused on the RIAE by revealing the vacuum of existing evidence for some RIAE and cancer types. According to the findings: healthcare resource use and costs were susceptive to study design and study population differences, which made observational cohort studies a more suitable source. Moreover, (updated) studies performed in the same countries as a source for input parameter improves validity. When these data are not available, the difference in health service price, clinical practice and environmental factors should be taken into consideration while introducing costs into economic evaluation.

## Figures and Tables

**Figure 1 cancers-14-02444-f001:**
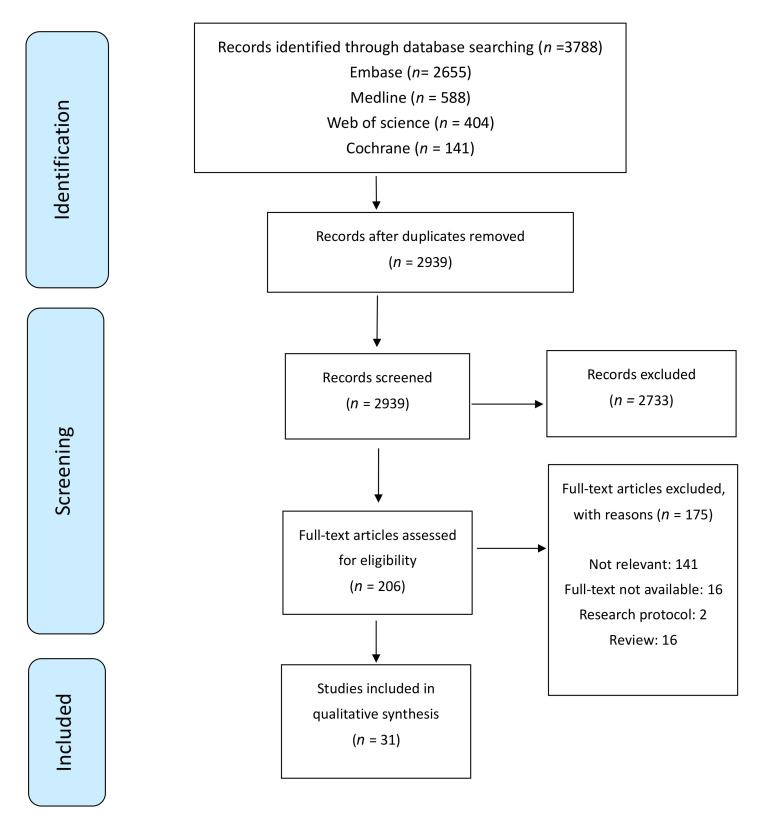
Flowchart for literature search process.

**Table 1 cancers-14-02444-t001:** General study characteristics of included literature.

Cancer Type	Country	Trial and Observational Study	Modeling Study
**Brain cancer**	U.S.	**0**	**1**
**Breast cancer**	U.S.	**1**	**2**
**Head and neck cancer**	Australia	1	0
	Germany	1	0
	Japan	2	0
	Netherlands	0	1
	Taiwan	1	0
	U.S.	8	0
**Medulloblastoma**	Japan	0	1
	Sweden	0	1
**Prostate cancer**	Ireland	1	0
	Netherlands	0	2
	U.S.	3	3
	UK	0	1
	Canada	0	1
	**Total**	**18**	**13**

**Table 2 cancers-14-02444-t002:** Radiotherapy-induced adverse effects studied in included literature.

Cancer Type	Radiotherapy-Induced Adverse Effects	Trials/Observational Studies (N = 4/14)	Modeling Study (N = 13)
**Breast Cancer**	Acute skin toxicity	1	0
	Cardiac adverse event	0	2
	Acute complication	0	1
	Contralateral breast cancer	0	1
	Lung cancer	0	1
**Head and neck cancer**	Craniofacial osteoradionecrosis	1	0
	Dysphagia	2	1
	Oral complications/mucositis	8	0
	Pharynx hypomotility	1	0
	Pneumonia	1	0
	Radiation-induced diarrhea	1	0
**Prostate cancer**	GI toxicity	1 ^1^	7 ^1^
	GU toxicity	1 ^1^	3 ^1^
	Radiation cystitis	1	0
	Urinary toxicity and rectal toxicity	1	0
	Sexual dysfunction toxicity	0	1
**Brain cancer and Medulloblastoma**	Growth hormone deficiency	0	2 ^1^
	Hearing loss	0	2 ^1^
	Hypothyroidism	0	1 ^1^
	Osteoporosis	0	1 ^1^

Abbreviations: GI: gastrointestinal, GU: genitourinary. ^1^ Some studies report more than one toxicity.

**Table 3 cancers-14-02444-t003:** Reported healthcare resource use converted into monetary value ^1^.

Toxicity	Hospitalization (Day)	Outpatient Visit	Others	Cost ($)	Perspective	Ref./Country
**Head and neck cancer**						
Dysphagia	4			2719	payer	Altman et al., 2010 [34]/US
Dysphagia	4.5		Nutritional supplements (days): 38.6; gastrostomy: 0.42	3486	payer	Bennett et al., 2001 [35]/DE
Mucositis		Nurse visit: 7.7; physician visit: 3; nutritionist visit: 3	Nutritional supplements (cans): 235.2	3165	payer	Peterman et al., 2001 [33]/US
Mucositis				4291 ^2^	payer	Elting et al., 2007 [31]/US
Grades 1 and 2	0.5	Dietician visits: 3; dental oncologist visit: 1.6	Gastrostomy tube (day): 6.1; opioid use (d): 23	4878		
Grades 3 and 4	1	Dietician visits: 3.8; dental oncologist visit: 2.3	Gastrostomy tube (day): 7.2; opioid use (d): 29	5198		
Mucositis	47		Opioid administration (mg): 6478	15,986	payer	Kubota et al., 2015 [36]/JP
Mucositis			Nutritional supplements (day): 27; opioid administration (mg): 3959	744	payer	Tsujimoto et al., 2015 [37]/JP
Mouth and throat soreness				1451 ^3^	payer	Murphy et al., 2009 [32]/US
Maximum pain score: 1	1		Nutritional Visits: 1.6	918		
Maximum pain score: 2	1.2		Nutritional Visits: 1.8	1083		
Maximum pain score: 3	1.6		Nutritional Visits: 2.4	1444		
Maximum pain score: 4	2.3		Nutritional Visits: 3.6	2099		
Oral complications, dehydration/electrolyte imbalance, infection and fever, malaise/fatigue	10.2			5073	payer	Lang et al., 2009 [38]/US
Pneumonia	5			3127	payer	Chu et al., 2013 [39]/TW
Pharynx hypomotility	13			5940	payer	Delaney et al., 1995 [40]/AU
Radiation-induced diarrhea	1.3	Physician visit: 0.33	Gastrostomy: 0.28	559	payer	Sonis et al., 2015 [41]/US

Abbreviations: US: United States, DE: Germany, JP: Japan, TW: Taiwan, AU: Australia. ^1^ The reported healthcare resource use related to RIAE was obtained and converted to monetary terms using the National Health Service (NHS) price set (2016–2017). All costs were translated to U.S. dollars with the annual average ratio of the publishing year of the study. The costs were adjusted for inflation to 2020 with CPI rate of the country where the study took place. ^2^ Calculated with assumption as followed: 15% patient with grade 0; 26% patient with grade 1 or 2; 59% patient with grade 3 or 4. ^3^ Calculated with assumption as followed: 6% patient with pain score 1; 15% patient with pain score 2; 51% patient with pain score 3; 25% patient with pain score 4. Pain score estimated with mouth and throat soreness [42].

**Table 4 cancers-14-02444-t004:** Reported healthcare costs ^1^.

Toxicity	Hospitalized	Outpatient Visit	Lab Test	Medication	Others	Cost ($)	Perspective/Cost Type	Ref./Country
**Head and neck cancer**								
Oral complications	+				+	7507	payer/institutional	Lang et al., 2009 [38]/US
Mucositis	+	+	+	+	+	20,728	payer/institutional	Nonzee et al., 2008 [45]/US
Mucositis	+	+	+	+	+	4312 (reimburse)/5903(charge)	payer/institutional and charging	Peterman et al., 2001 [33]/US
Mucositis	+	+		+	+ (Nutritionist, medication)	7462 ^3^	payer/institutional	Elting et al., 2007 [31]/US
Grades 1 and 2						2122 ^4^		
Grades 3 and 4						4493 ^4^		
Osteoradionecrosis	+			+	+ (hyperbaric oxygen, surgical debridement, simultaneous resection–microvascular free flap reconstruction)	66,399	payer/reimbursement	Kelishadi et al., 2009 [46]/US
Pneumonia	+					2014	payer/reimbursement	Chu et al., 2013 [39]/TW
**Prostate cancer**								
GU toxicity or GI toxicity	+	+	+	+	+	1314	payer/institutional	Redmond et al., 2018 [43]/IE
GU toxicity or GI toxicity				+	+	1352 ^2^	payer/reimbursement	Voong et al., 2017 [44]/US
Grade ≤ 1 rectal toxicity and grade ≤ 1 urinary toxicity						684		
Grade ≤ 1 rectal toxicity and grade 2,3 urinary toxicity						1284		
Grade 2, 3 rectal toxicity and grade ≤ 1 urinary toxicity						1623		
Grade 2, 3 rectal toxicity and grade 2, 3 urinary toxicity						8165		
Radiation cystitis	+				+	1,056,443	payer/institutional	Kiechle et al., 2016 [47]/US
**Breast cancer**								
Acute skin toxicity					+	149	healthcare sector/out-of-pocket	Schnur et al., 2012 [29]/US

Abbreviations: GI: gastrointestinal, GU: genitourinary, US: United States, IE: Ireland, JP: Japan, TW: Taiwan. ^1^ All costs were translated to U.S. dollars with the annual average ratio of the published year of the study. The costs were adjusted for inflation to 2020 with CPI rate of the country where the study took place. ^2^ Calculated with assumptions as followed: 78% patient with grade ≤ 1 rectal toxicity and grade ≤ 1 urinary toxicity; 11% patient with grade ≤ 1 rectal toxicity and grade 2, 3 urinary toxicity; 7% patient with grade 2, 3 rectal toxicity and grade ≤ 1 urinary toxicity; 4% patient with grade 2, 3 rectal toxicity and grade 2, 3 urinary toxicity. The percentage is calculated from condition probability: % of rectal toxicity * % of urinary toxicity = % of having both rectal toxicity and urinary toxicity. ^3^ Calculated by subtracting the mean cost of RT alone among patients without mucositis from the mean cost among patients with mucositis. ^4^ Cost calculated by the increased use of resources among patients with mucositis compared to patients without mucositis.

**Table 5 cancers-14-02444-t005:** Healthcare costs reported by modeling studies.

Toxicity/Cost (USD)				Ref./Data Source/Country
**Head and neck cancer**				
Xerostomia				
Grade 0 to 1	Grade 2	Grade 3	Grade 4	
184	199			Ramaekers et al., 2012 [48]/EO/NL
Dysphagia				
Grade 0 to 1	Grade 2	Grade 3	Grade 4	
99	760			Ramaekers et al., 2012 [48]/EO/NL
**Prostate cancer**				
GI toxicity				
Grade 1	Grade 2	Grade 3	Grade 4	
288				Parthan et al., 2013 [49]/EO/US
	1140			Cooperberg et al., 2013 [50]/EO/US
386	571	4912		Vanneste et al., 2015 [51]/EO/NL
	674	5678		van Wijk et al., 2017 [52]/EO/NL
	563	2421		Yong et al., 2012 [22]/EO, ES/CA
	1054	2108		Hummel et al., 2012 [21]/EO/UK
	47	3286		Peters et al., 2016 [53]/EO/US
Rectal toxicity				
Grade 1	Grade 2	Grade 3	Grade 4	
395	1417	4791	13,637	Hutchinson et al., 2016 [54]/EO/US
GU toxicity				
Grade 1	Grade 2	Grade 3	Grade 4	
	1541			Cooperberg et al., 2013 [50]/EO/US
239				Parthan et al., 2013 [49]/EO/US
	234	4738		Peters et al., 2016 [53]/EO/US
**Medulloblastoma**				
Hearing loss	GHD	Hypothyroidism	Osteoporosis	
2360 ^1^				Hirano et al., 2014 [55]/TG/JP
5829 ^3^	15,542 (≤19 y/o); 1554 (>19 y/o) ^2^	132 ^3^	544 ^2^	Lundkvist et al., 2005 [30] ^5^/ES/SE
	17,773 (4 y/o); 35,208(9 y/o) ^4^			Mailhot Vega et al., 2015 [56]/TG/US
**Breast cancer**				
Acute complications		4245		Patel et al., 2017 [57]/EO/US
Major coronary event		29,322		Patel et al., 2017 [57]/EO/US
Cardiac Adverse Event (MI)		9167		Ward et al., 2019 [58]/TG/US
Lung cancer		54,132		Patel et al., 2017 [57]/EO/US
Contralateral breast cancer		11,847		Patel et al., 2017 [57]/EO/US

Abbreviations: EO: expert opinion, ES: empirical study, TG: treatment guideline; MI: myocardial infarction, GHD: growth hormone deficiency, GI: gastrointestinal, GU: genitourinary, NL: Netherlands, US: United States, CA: Canada, JP: Japan, UK: United Kingdom, SE: Sweden. ^1^ Included expense for a hearing test, hearing aid fitting test, annual hearing test for the following two years after hearing loss occurred; the hearing aid duration with five-year duration. ^2^ Annual costs for the remaining lifetime; ^3^ costs per year; ^4^ costs per year for patients from ages 4 to 9. ^5^ Societal perspective.

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
