# Peer review of "Medical Resource Use and Medical Costs for Radiotherapy-Related Adverse Effects: A Systematic Review"

_cancers, 2022, doi:10.3390/cancers14102444_

Round 1
Reviewer 1 Report
In this study the authors systematically reviewed literatures to identify the existing evidence of healthcare resource use and costs related to radiotherapy induced adverse effects. They also provided guidance for including resource uses and costs into economic evaluation.
Both literature survey and selection mechanisms look sound, the analysis method seems reasonable, and the results are described concisely both in the text and tables. It is considered the recommendations are practical and conclusions agreeable.
A few comments for improvement are:
- Front page: Place the affiliated institutions in the order of authors (note 2nd author’s institution is in 3rd).
- L41-43: “Several studies showed that patients receiving the innovative radiotherapy treatments have significantly lower risk of adverse effects compared to conventional treatments.” => References are needed.
- L51-53: “Several studies reported cost savings for avoiding RIAE, but a systematic review that summarizes the currently available evidence is currently lacking.” => References are needed.
- L173-174: “Non-modelling-based studies for brain cancer, breast cancer nor eye cancer were scarce or unavailable.” => Check if grammar is correct.
- Table 3: Widen the width of cost column so “15986” can be listed in a single row. Describe the rule used for cost calculation on the bottom of the table so readers can understand it even with reading the table only.
- Table 4: Place “Osteoradionecrosis” below the last “Mucositis.” Describe the rule used for cost adjustment on the bottom of the table.
- L253: “… to hearing lost is …” => “… to hearing loss is …”
- L295: “consequently …” => “Consequently …”
- All of tables: How about to add more information (columns) such as country and year range of patient care provided. As brought up by yourselves in the discussion such information would be very useful in understanding local healthcare situations/conditions.
Reviewer 2 Report
Interesting results; in the everyday practice it is very difficult to exactly calculate the exact price of treatment of complications which might be very different according to each country health policy
Author Response
Dear reviewer,
Thank you for your positive comments.
We are glad to contribute to this research area.
Best wishes,
YH Chen
Reviewer 3 Report
Please find my review report for cancers-1710385. Thank you for considering me as one of the reviewer.
